# J3ExoA: A Novel Anti-HIV Immunotoxin Fusion of Anti-Gp120 J3VHH and PE38 Fragment of *Pseudomonas* Exotoxin A

**DOI:** 10.3390/ph18091305

**Published:** 2025-08-30

**Authors:** Seth H. Pincus, Kun Luo, Tami Peters, James T. Gordy, Frances M. Cole, Grant Klug, Kelli Ober, Tamera K. Marcotte, Richard B. Markham

**Affiliations:** 1Department of Chemistry and Biochemistry, Montana State University, Bozeman, MT 59715, USA; 2Department of Molecular Microbiology and Immunology, Johns Hopkins Bloomberg School of Public Health, Baltimore, MD 21205, USAjgordy2@jhu.edu (J.T.G.); 3Animal Resources Center, Montana State University, Bozeman, MT 59715, USA

**Keywords:** HIV reservoir eradication, cytotoxic immunoconjugate, HIV envelope protein

## Abstract

**Background.** We are developing cytotoxic anti-HIV immunoconjugates to attack the reservoir of infected cells that persist after years of fully suppressive anti-retroviral therapy. **Methods.** We have produced a chimeric fusion protein, J3ExoA, consisting of J3VHH, a broadly reactive anti-gp120 camelid nanobody, joined to the de-immunized PE38 fragment of *Pseudomonas* exotoxin A. The efficacy of J3ExoA was compared to that of a well-studied anti-gp41 immunotoxin (IT), 7B2-dgA, in cytotoxicity assays and for inhibition of infectivity. Immunogenicity of the ITs was tested in mice. **Results.** J3ExoA killed cells expressing the HIV envelope with specificity in concentrations in the ng/mL range. Of all anti-HIV ITs we have tested, only J3ExoA compared to 7B2-dgA in cytotoxic efficacy, although there were differences between the two ITs on different target cells. J3ExoA suppressed the spread of HIV infection in tissue culture. J3ExoA was less immunogenic than 7B2-dgA, but mice made antibodies to both portions of the fusion protein. **Conclusions.** J3ExoA represents a novel IT that may be used to eliminate infected cells in the persistent HIV reservoir of infection, the barrier to an HIV “cure.” Additional approaches for addressing IT immunogenicity are discussed.

## 1. Introduction

Although HIV infection can be effectively suppressed and AIDS avoided by use of combination anti-retroviral therapy (ART), infection cannot be eradicated [1,2]. Even after years of fully suppressive therapy, discontinuation of ART leads to a rapid rebound in viremia. Long-lived memory CD4 T-cells represent much of the reservoir [1], with cells of the myeloid lineage likely also contributing. HIV protein expression in infected cells during ART is either fully latent or expressed at very low levels [3]. Latency disrupting agents (LDAs) are being developed to increase expression of virus proteins [4,5,6,7,8], for so called “activate and purge” protocols. We and others have developed cytotoxic immunoconjugates to eliminate the cells activated by these LDAs to express HIV proteins, including immunotoxins (ITs), antibody-drug conjugates (ADCs) and radioimmunoconjugates. These have been tested in a variety of in vitro systems and found to be highly efficacious in killing HIV-infected cells and limiting the spread of infection. Therapeutic effects have been observed in mouse and macaque models of HIV [9,10,11,12,13,14,15,16,17,18,19,20,21,22,23,24].

The HIV envelope protein (Env or gp160, consisting of an external domain gp120, and a transmembrane domain gp41) is the sole viral protein expressed on the virion surface or on HIV-infected cells, and is therefore our target. We have tested a variety of different full-length antibodies for their ability to deliver an IT and found the human anti-gp41 monoclonal antibody (mAb) 7B2 to be the most effective [11,12,25]. VHH nanobodies are single-domain peptides of camelid origin. VHH nanobodies consist of a single heavy chain variable region (hence, VHH) of ~15 KD capable of binding antigen with high affinity, and because of their small size are able to penetrate to epitopes inaccessible to full-sized Abs [26]. J3VHH has been identified as a broadly-reactive neutralizing anti-gp120 nanobody [27]. J3VHH neutralizes HIV by binding to and blocking the CD4-binding site on gp120 [28]

Cytotoxic immunoconjugates are bifunctional molecules used to kill cells expressing a target antigen, HIV Env in this case. One domain consists of the targeting molecule, either an Ab or a binding partner of the target. The other domain is the toxic moiety: if a protein toxin is used, we term it an IT; the term ADC is applied if a low MW cytotoxic drug is conjugated, or radioimmunoconjugate if a high energy radioisotope is bound to the antibody. All these forms of anti-HIV immunoconjugates have been tested in vitro and/or in animal models of HIV infection [9,13,14,22,24]. We have shown efficacy in mice and macaques of an IT consisting of mAb 7B2 chemically conjugated to the deglycosylated A chain of the plant toxin ricin (dgA) [13,14]. However, the utility of 7B2-dgA was found to be limited due to its immunogenicity, resulting in loss of efficacy within 2-3 weeks, a well-noted problem of ITs.

Seeking to produce an effective anti-HIV IT with limited immunogenicity, we fused J3VHH to a fragment of *Pseudomonas* exotoxin A that has been engineered to eliminate B and T cell epitopes yet retains toxic activity. J3VHH was chosen because of its high affinity binding to gp120, broad and potent neutralizing activity, and it exists as a single chain amenable to genetic manipulation. We have compared J3ExoA to 7B2-dgA for cytotoxic efficacy, ability to inhibit infection, and for immunogenicity in mice. J3ExoA demonstrated excellent in vitro effectiveness but retained immunogenicity.

## 2. Results

### 2.1. Design and Production of J3ExoA

J3VHH is a broadly neutralizing anti-gp120 antibody isolated from a phagemid library derived from a llama hyperimmunized with a mixture of trimeric gp140 molecules [27]. ExoA consists of the PE38 fragment of *Pseudomonas* exotoxin A that has been de-immunized by alanine substitution within key epitopes [29,30]. An N-terminal His-tag has been added for purification, followed by a leader sequence and the J3VHH sequence. J3 is joined to the C-terminal ExoA by a linker region containing a central Myc-tag located between two flexible peptides. Figure 1A depicts a stick figure of J3ExoA color-coded to the amino acid sequence shown beneath. Figure 1B shows reducing and non-reducing SDS-PAGE gels of J3ExoA and J3VHH. The majority product is of the correct MW (J3VHH 17 KD, J3ExoA 60 KD), with the non-reducing gel showing some degree of aggregation. A minor amount of protein degradation was observed in J3ExoA, and much less so in J3VHH.

### 2.2. Comparative Cytotoxicity of J3ExoA and 7B2-dgA on Cell Lines

The cytotoxicity and specificity of IT killing were tested on three sets of HIV Env-expressing cell lines and their Env-negative parental cells. Two sets of cells were transfected to stably express gp160 in its native conformation, whereas one set, H9/NL4-3, represents a persistently-infected cell line continuously producing infectious HIV. Figure 2 demonstrates the expression of the HIV envelope on the surfaces of the different cell lines used in these experiments, as detected by anti-gp120 mAb PGT126 or anti-gp41 mAb 7B2 + sCD4_183_ and FITC-conjugated anti-human IgG. The addition of sCD4 to gp41-targeting ITs enhances their activity 10–30X [10,31] by increasing both antigen exposure on the cell surface and internalization. Thus, assays described below utilizing 7B2-dgA were performed in the presence of sCD4. Neither of the antibodies bound the non-expressing parental cells 293T, CHO, and H9. The two cell lines derived from 293T, 92UG, and C97 had notably different expression patterns. The 92UG cells showed high expression of both HIV Clade A gp120 and gp41, whereas the Clade C-derived C97 appeared to contain three populations of cells: 1. Env-negative, 2. Low level expression of both gp120 and gp41, and 3. Moderate level expression of gp120, but low level gp41. Because these experiments were performed by indirect immunofluorescence, we cannot confirm the homogeneity of these populations by 2D fluorescence. The CHO.Env15 cells also appeared to contain the same three populations as C97, but in the third population both gp120 and gp41 demonstrated moderate levels of cell surface expression. The persistently infected H9/NL4-3 had high level expression of gp41, but very low levels of binding by PGT126.

Cytotoxicity testing was performed by MTS dye reduction, a measure of oxidative phosphorylation and consequently of the number of metabolically active cells. Figure 3 compares the cytotoxic efficacy of J3ExoA to our best ricin-based IT, 7B2-dgA. For all cytotoxicity assays CD4-IgG2 500 ng/mL was added to cultures containing 7B2-dgA. In Figure 3A, the panel of cells based on 293T were tested. The 92UG cells, which demonstrated high levels of both gp120 and gp41 on the cell surface, were most sensitive to the ITs, with both demonstrating an IC_50_ of 0.65 ng/mL. The C97 cells showed a more complicated pattern, with neither IT obtaining an IC_50_ at concentrations < 2 µg/mL, equivalent to what was observed as non-specific toxicity on Env-negative 293T. However, J3ExoA produced consistent killing of ~30–40% of C97 at concentrations > 10 ng/mL. This likely represents the cell population observed to express moderate levels of gp120 by indirect immunofluorescence (Figure 2A). Notably, 7B2-dgA was completely ineffective on C97 until reaching the stage of non-specific toxicity. These data indicate that there was a minimal level of Env expression necessary to obtain a cytotoxic effect. Figure 3B evaluates cytotoxicity of the ITs on the Env-negative cell lines C8166.R5, which were used in infectivity assays, and H9, the parent of H9/NL4-3. J3ExoA failed to demonstrate non-specific cytotoxicity at concentrations as high as 30 µg/mL; 7B2-dgA demonstrated greater toxicity with IC_50_’s of 7 and 20 µg/mL for C8166.R5 and H9, respectively. The failure of the ITs to kill Env-negative cells at concentrations orders of magnitude higher than those demonstrating killing on Env-expressing cells is evidence of the specificity of IT killing.

Figure 4A demonstrates the need for the ExoA portion of the molecule to obtain killing by J3ExoA with the demonstration that J3VHH failed to kill 92UG cells. This experiment also confirmed the extreme sensitivity of these cells to both 7B2-dgA and J3-ExoA, and that the addition of CD4-IgG2 had no effect on killing by J3ExoA, with virtually superimposed curves with or without CD4-IgG2. Figure 4B shows that with 92UG cells, each IT had a therapeutic ratio (IC_50_ on Env-negative cells/IC_50_ on Env+ cells) > 1000. Cytotoxicity of the ITs was also studied on CHO derived cells (Figure 5A). The IC_50_ on CHO.Env15 cells was 4.4 and 40 ng/mL for 7B2-dgA and J3ExoA respectively, whereas the IC_50_ on CHO.psv was 3–5 µg/mL. Neither IT achieved 100% killing, consistent with the observations by immunofluorescence that not all cells express Env on the cell surface (Figure 2B). H9/NL4-3 cells maintain a continuous productive infection with HIV, thus differing from 92UG, C97, and CHO.Env15, which were transfected with molecular clones expressing Env only. H9/NL4-3 also differed in that expression of gp41 was markedly greater than that of gp120. Cytotoxicity of both ITs was compared on H9/NL4-3 and 92UG cells (Figure 5B). Whereas both ITs were equally effective on 92UG, only 7B2-dgA killed the H9/NL4-3 cells, consistent with the lower gp120 expression on the H9/NL4-3 cells.

### 2.3. Ability of ITs to Inhibit the Spread of Tissue Culture Infection

Initial experiments were performed in phytohemagglutinin (PHA)-induced blast cultures and compared the antiviral effect of J3VHH to that of J3ExoA (Figure 6). First, 1 × 10^8^ uninfected peripheral blood mononuclear cells were treated with PHA for 2 days in R10 media and then incubated overnight with HIV_BAL_ to initiate infection. The following morning, 50 nM J3VHH, J3ExoA, or control IgG were added to the culture. The production of p24 was measured 5 days following infection (Figure 6A). Treatment with J3ExoA resulted in a marked, but not complete, reduction in p24 production (*p* < 0.01). J3VHH produced a non-significant decrease in p24 production. Because J3VHH has been reported to be a highly effective neutralizing antibody, the lack of effect could suggest that this assay measures the direct cell-to-cell spread of infection, rather than infection by cell-free virus. To demonstrate that the antiviral effect was not the result of neutralizing virus and inhibiting its binding to target cells, we used a single cycle infection of MAGI-CCR5 cells to assess the ability of J3VHH and J3ExoA to neutralize infection. The results showed J3VHH outperforming J3ExoA at every concentration tested (Figure 6B). These data suggest that the genetic modifications of J3ExoA likely result in decreased affinity of binding compared to the native VHH. Therefore, the antiviral efficacy of J3ExoA in the assay measuring spread of infection (Figure 6A) cannot be explained by blocking of virus binding to target cell by the J3 moiety.

To further test the anti-viral efficacy of the ITs we used C8166.R5 lymphoma cells, which have been shown to be functionally equivalent to PHA blasts for growing HIV and testing antiviral agents [32]. Figure 7 shows the effects on the spread of infection (HIV_JR-CSF_) when ITs or Ab were added at 50 ng/mL. When J3ExoA, J3VHH, and 7B2-dgA were compared to untreated cells, only J3ExoA produced a significant effect at all time points tested. In this experiment sCD4 was omitted from the 7B2-dgA, because the neutralization of infectivity by sCD4 hindered analysis of the IT effect.

### 2.4. Immunogenicity and Toxicity of J3ExoA

Immunogenicity is the major factor limiting the utility of ITs [13,29,33]. J3ExoA was designed to minimize immunogenicity eliminating epitopes by alanine substitution of key AAs within epitopes [30]. We compared the immunogenicity of J3ExoA to that of 7B2-dgA by repeatedly injecting immunocompetent, uninfected mice intraperitoneally with 10 µg/mouse of IT, a high therapeutic dose [14], over a period of three months. Each IT carried a xenogeneic antibody linked to a partial toxin molecule. Figure 8 shows that anti-IT IgG to J3ExoA was slower to develop than to 7B2-dgA, but that ultimately both ITs elicited comparative levels of anti-IT antibody. We next sought to determine which portion of J3ExoA elicited an antibody response (Figure 9). Equivalent levels of antibody were observed to both the J3VHH and the toxin. It was also noted that there was no evidence of toxicity in these animals, thus demonstrating safety at a high (~400 µg/kg), repetitive dose.

## 3. Discussion

In this manuscript we describe a novel anti-HIV IT, J3ExoA, a fusion protein with an N-terminal domain derived from the broadly neutralizing anti-gp120 camelid VHH J3 [27], fused via a linker region to PE38, which contains the toxic domain of *Pseudomonas* exotoxin A. The toxic domain was further deimmunized by alanine substitution [29,30]. J3ExoA was compared to the previously described anti-HIV IT 7B2-dgA [9,12,13]. We tested in vitro for cytotoxicity, specificity of killing, and inhibition of the spread of infection, and in mice for immunogenicity and toxicity. We have previously compared many ITs that target anti-gp160 epitopes distinctly different from those targeted by 7B2 and found them lacking [11,12]. In the present study, we demonstrate that J3ExoA favorably compares to 7B2-dgA + sCD4 in terms of cytotoxicity (Figure 3, Figure 4 and Figure 5), lack of non-specific killing, and inhibition of the spread of infection (Figure 6 and Figure 7). The need to add sCD4 to 7B2-dgA brings practical challenges to that treatment regimen. In mice, neither IT demonstrated toxicity; both were immunogenic (Figure 8 and Figure 9).

Cytotoxicity was measured on four different target cells, all of which express HIV Env, as well as on their Env-negative parental cells. As can be observed in Figure 2, some of the cell lines consisted of a single Gaussian population of cells, while others contained multiple cell populations, each expressing different levels of gp120 and gp41. Another difference was that H9/NL4-3 constitutively produced infectious virus, whereas the others were transfected to express Env. Differences between the two ITs in cytotoxicity on different cell lines yields useful information regarding amount of cell surface necessary to initiate cytotoxicity. For example, C97 demonstrated two different patterns of gp41 expression (which may be termed none and low, when compared to the Env-negative 293T parent), and three levels of gp120 expression (none, low and high). When we measure cytotoxicity of the ITs on C97 (Figure 3A), we observed no cytotoxicity of the anti-gp41 IT until we reached a high concentration that approximated the concentration killing the Env-negative parent. In contrast, J3ExoA killed ~40% of C97 at very low concentrations, corresponding in size to the population of Env-high cells observed by immunofluorescence. A different cell line, CHO.Env15, expressed equivalent levels of gp120 and gp41 on the cell surface containing three populations of cells (Figure 2B). While both ITs achieved killing on CHO.ENV15 at concentrations orders of magnitude less than on the Env-negative CHO parent, 7B2-dgA +sCD4 produced significant killing at concentrations ten-fold lower than J3ExoA (Figure 5A). Neither IT killed > 60% of the CHO-Env15 cells, again corresponding to the population of cells expressing Env by fluorescence. Differences in killing by the ITs on the persistently-infected H9/NL4-3 can be attributed to the level of gp120 vs. gp41 expressed by those cells. We demonstrated the specificity of killing by J3ExoA by showing that cytotoxicity on Env-negative cell lines required markedly higher concentrations of IT than on Env+ cells (Figure 3, Figure 4 and Figure 5) and by the failure of J3VHH to kill Env+ cells (Figure 4). Non-specific cytotoxicity of J3ExoA on Env-negative cells was less than that observed for 7B2-dgA.

Measuring the ability of the ITs to inhibit the spread of infection proves more difficult when the treatment includes a moiety that alone has been shown to neutralize HIV. For this reason, we excluded sCD4 when using 7B2-dgA in the experiment shown in Figure 7. For J3ExoA, we included J3VHH as a control. The efficacy of the ITs was tested in both primary human T-cell blasts (Figure 6) and C8166.R5 human CD4+ T-cell lymphoma cells (Figure 7). Infection of C8166.R5 cells resulted in no detection of p24 antigen until 24 h followed by an explosive growth of infection between 30 and 60 h (see p24 production in untreated cells in Figure 7). At the same time, cells demonstrated viral cytopathic effects including blebbing and syncytium formation, before massive cell death was noted by days four to five. Antiviral effects were most clearly demonstrated earlier in this infectious cycle. For example, J3VHH provided significant neutralization only at the earliest time point, 38 h. In contrast, J3ExoA significantly inhibited the spread of infection, when compared to the untreated control, at all time points tested.

Using an aggressive immunization protocol and ITs made in research laboratories, our studies have demonstrated immunogenicity of both components of the J3ExoA complex. The degree to which this would represent a problem for clinical application would depend, in part, on how many applications might be required to achieve clinical benefit. The production of anti-IT Abs generally leads to more rapid clearance, resulting in loss of IT efficacy, as we observed in IT-treated SHIV-infected macaques [13]. Different approaches can be taken to mitigate this effect. For any clinical use, standards for manufacture would be optimized and more rigorous, and thus less likely to contain materials such as molecular aggregates that might enhance immunogenicity. General approaches to minimizing immunogenicity include genetic modification of either component of the IT, PEGylation [13,34], and immunosuppression [35,36,37], although the latter may be unwise in HIV-infected patients. In seeking to mitigate the immune response to either the monoclonal antibody or VHH, efforts have focused on humanization of the antibody/VHH component [38,39,40,41,42,43,44,45]. Particularly encouraging are results demonstrating sustained efficacy of anti-tumor necrosis factor humanized VHH in the treatment of rheumatoid arthritis [46]. The use of VHH was also associated with reduction in the theoretical risk of immune complex associated inflammatory responses, attributable to the reduction, with use of VHH, in Fc-mediated engagement of granulocytes [47]. Both targeting domains tested here, J3 and 7B2, were fully xenogeneic for the mice.

J3ExoA was designed by genetically removing B and T cell epitopes within the toxic moiety, as defined by the work of Pastan’s group [29,30,48]. We found that the modified ExoA retained immunogenicity, given the caveats noted above. In our experiments we did not compare the genetically modified form to the native form to determine the degree to which the modifications affected immunogenicity. Because there is obvious evolutionary advantage to developing immune and inflammatory responses that lead to the rapid elimination of plant and bacterial toxins, these molecules are highly immunogenic and proinflammatory, and further genetic modifications may lead to loss of the toxin activity. Should genetic modification prove insufficient, the sequential use of a panel of toxins, including the ricin and ExoA employed in this study, as well as diphtheria toxin containing ITs [49,50], might blunt the anti-toxin effect. A conventional approach to avoiding the immunogenicity of ITs is the use of ADCs and radioimmunoconjugates; and we have done so for HIV. We have made anti-HIV ADCs [9,13] and radio-immunoconjugates (A.-S. Kuhlmann, et al. under review), all based on antibody 7B2. Strikingly, we found there were significant differences in the mode and kinetics of killing of each form of the cytotoxic immunoconjugate even when tested on the same target cells (A.-S. Kuhlmann et al., under review; S.H. Pincus et al., under review), and that ITs were more potent, acted more rapidly and had fewer non-specific cytotoxic effects on bystander cells. Thus, despite issues of immunogenicity, there is continued reason for the development of ITs such as J3ExoA. 

We propose that these immunoconjugates would be utilized in an “activate and purge” clinical protocol to eradicate the persistent reservoir of HIV infection. Patients whose viremia is fully suppressed by ART would be candidates. ART would be continued or even intensified during the activation phase, to prevent infection of uninfected lymphocytes. A number of agents have demonstrated their ability, with varying degrees of efficacy, to activate HIV expression [4,5,6,7,8], but lacking an effective means for purging the activated cells, produced little clinical effect. We propose that once latently-infected cells have been activated to express HIV envelope on the cell surface, the cytotoxic immunoconjugates would be administered, specifically killing the Env-expressing cells. A key unknown in this process is whether LDAs can induce a sufficient level of Env expression to result in IT-mediated cytotoxicity. The next step in developing this approach will be to test the efficacy and toxicity of the cytotoxic immunoconjugates in macaques infected with the chimeric simian-human Env immunodeficiency virus [13] that are ART-suppressed and treated with LDAs.

## 4. Materials and Methods

### 4.1. Production of J3ExoA and J3VHH

The synthetic gene encoding the J3VHH was constructed by Genscript (Piscataway, NJ, USA) utilizing the sequence from McCoy et al. [27], flanked by EcoRI and XhoI sites. Utilizing standard cloning techniques, J3VHH was ligated into the *E. coli* expression vector pET-47b(+) via EcoRI and XhoI sites. Our laboratory previously constructed a pET-47b(+) vector with the *Pseudomonas aeruginosa* gene encoding the Exotoxin A subunit, described in Geoghegan et al. [51]. The construct encoding the fusion product of J3VHH-*P. aeruginosa* Exotoxin A (J3ExoA) was created by ligating the J3VHH region upstream of the ExoA construct by EcoRI and XhoI sites. Both J3VHH and J3ExoA in pET-47b(+) were transformed into competent BL21 DE3 cells (New England Biolabs, Ipswich, MA, USA). Expression, purification, and refolding of J3VHH and J3ExoA followed the protocol of Brinkmann [52]. One L cultures of *E. coli* were grown in LB containing 50 µg/mL kanamycin to OD_600_ 0.6 and induced with 1 mM IPTG (Lab Scientific Inc., Highlands, NJ, USA) for 3 h at 37°. Cells were pelleted, washed, and lysed in the presence of 6M guanidine HCl. Ni-NTA Agarose (QIAGEN, Valencia, CA, USA) was added to the lysate. After washing, the His-tagged J3ExoA or J3VHH was eluted with 8 M urea, 250 mM imidazole, 50 mM NaH_2_PO_4_, 500 mM NaCl, 300 mM DTT) and diluted 1:100 into refolding buffer (100mM Tris, 500 mM L-arginine, 8 mM oxidized glutathione, 2 mM EDTA) and incubated overnight at 10°. The final refolded proteins were concentrated and buffer exchanged into PBS using a 10 KD cutoff centrifugal filter (Amicon, EMD Millipore, Billerica, MA, USA). Protein concentration was determined using Bradford assay and purity assessed by SDS-PAGE (Figure 1). J3VHH and J3ExoA were stored in aliquots at 2–5 mg/mL stored at −80°.

### 4.2. Reagents and Cell Lines

The provenance and characteristics of the cell lines used in these studies have been described in detail elsewhere. Cell lines 293T/92UG (henceforth 92UG) and 293T/C97 (C97) stably express native Env of HIV isolates clade A 92UG037.8 and clade C C97ZA012 in HEK-293T cells [53] and were the kind gift of Bing Chen (Boston Children’s Hospital, MA, USA). CHO.Env15 expresses the Env of HIV strain HTLV-III_B_ in Chinese hamster ovary (CHO) cells [54]. CHO.Env15 and its Env-negative parent CHO.psv were the gift of Ed Berger (NIAID, Bethesda MD, USA). H9/NL4-3 is a CD4+ T-cell lymphoma line persistently infected with the NL4-3 infectious molecular clone of HIV [12,15] and was the gift of Kathy Wehrley (NIAID Rocky Mountain Labs, Hamilton, MT, USA). The NL4-3 virus utilizes the Env derived from the HTLV-III_B_ isolate of HIV [55]. C8166.R5 cell is a CD4+ CXCR4+ lymphoma cell line transfected to express CCR5 and has been found non-inferior to PBMC blast cells for cultivating HIV and detecting anti-viral effects [32]. All cells were maintained in medium supplemented with 10% fetal bovine serum (FBS): cells based on HEK-293T and CHO were grown in Dulbecco’s Modified Essential Medium (4.5 g glucose/L), the lymphoma cell lines in RPMI 1640 (all from Life Technologies, Carlsbad, CA, USA), and incubated at 37 °C in a humidified 5% CO_2_ atmosphere.

Human IgG1 anti-HIV Env mAb 7B2 (anti-gp41) was created by James Robinson (Tulane University, New Orleans LA, USA); the antibody used in these studies was a gift from Bart Haynes (Duke University, Durham NC, USA) [56]. Anti-gp120 mAb PGT126 [57] was produced in our laboratory by transient transfection of HEK-293T cells using 293-Fectin reagent (Life Technologies) with plasmids (pcDNA3.1) containing synthetic constructs of the appropriate H and L chains. Transfected cells were grown in DMEM-medium with 5% fetal calf serum depleted of IgG by passage over Protein G agarose. Purified mAb was obtained by passage of supernatant over protein G agarose, acid elution, neutralization and buffer exchange into PBS. 7B2-dgA is an immunotoxin consisting of 7B2 conjugated to deglycosylated ricin A chain (dgA) with a disulfide-containing linker. Its construction and activity have been described elsewhere [9,12,13]. Two different forms of soluble CD4 (sCD4) were used in these assays; sCD4_183_ a monomer based on the N-terminal 183 amino acids of CD4 and CD4-IgG2 a tetramer in which the amino terminal domain replaces the V-regions of an IgG2 human antibody [58,59]. We utilized sCD4_183_ for indirect immunoassays based on detection of human Fc, and was obtained from Upjohn Laboratories (Kalamazoo, MI USA). CD4-IgG2 (also known as Pro542) was used in cytotoxicity assays and was a gift of Progenics Pharmaceuticals (Tarrytown, NY USA). FITC conjugated goat anti-human IgG and alkaline phosphatase-conjugated goat anti-mouse IgG were obtained from Novus Biologicals (Centennial, CO USA). Purified *Pseudomonas* exotoxin PE24 fragment was used as antigen in immunoassays. The protein (AA 51 to 269 of the intact toxin, Genbank HEC0566463.1) was expressed in *E.coli*, purified, and characterized at Gene Universal (Newark, DE, USA).

### 4.3. Immunofluorescence and Flow Cytometry

Fifty µL of primary mAb diluted to 20 µg/mL in PBA (PBS/1% Bovine Serum Albumin [Sigma Chemical, St. Louis, MO USA]/0.1% Na azide [Sigma]) were added to wells in 96-well round bottom plates (Costar, Lowell MA USA). Fifty µL of cells (10^6^ viable cells/mL in PBA) were then added and the plate incubated at 4 °C for 1 h. One hundred µL of PBS was added to the wells, the plates centrifuged and washed 2X in PBS. One hundred µL of FITC-conjugated anti human IgG (Novus Biologicals, Centennial, CO, USA, 2 µg/mL in PBA) were added to the cell pellet and the cells mixed. The cells were again incubated at 4 °C for 1 h, then washed twice and fixed in 2% paraformaldehyde overnight. Cells (1–3 × 10^4^), gated by forward and side scatter, were analyzed on an Accuri C6 flow cytometer (BD Biosciences, San Jose, CA, USA). Data was analyzed using FlowJo Software v11 (Treestar/BD Biosciences).

### 4.4. Cytotoxicity Assays

The MTS dye reduction assay, a measure of oxidative phosphorylation, was used to determine the cytotoxicity of the ITs [9,10,12]. Target cells, 1–2 × 10^4^ per well depending upon the cell type, were suspended in 0.2 mL in RPMI 1640 medium + 10% FBS, containing the indicated concentration of IT ± CD4-IgG2, and placed into flat bottom 96 well tissue culture plates. Samples were run in triplicate. Cells were incubated for 72 h, with MTS/PMS substrate (CellTiter Aqueous, Promega, Madison, WI, USA) added for the final three hours. Absorbance (A) was read at 490 nM on a microplate reader (BioTek, Winooski, VT, USA). Percent cytotoxicity was calculated according to the formula:{1 − [(A_IT_ − A_no cells_)/(A_no IT_ − A_no cells_)} × 100(1)

### 4.5. Infection Assays

Infection assays were performed in the C8166.R5 tissue culture cell line and in PHA activated peripheral blood mononuclear cells (PHA blasts). The outcome was read as production of p24 antigen. C8166.R5 assays utilized the R5-tropic HIV isolate JR-CSF (HIV Reagent Program, BEI Resources, Manassas VA, USA). Upon receipt of the virus, a stock was grown in PHA blasts, aliquoted, frozen at −80 °C and titered in C8166.R5. At the initiation of the experiment uninfected C8166.R5 cells (4 × 10^5^ per mL) were mixed with a predetermined dilution of virus stock, mixed, and added to flat bottom wells of 48-well plates containing the indicated ITs to a final volume of 1 mL of RPMI 1640 medium + 10% FBS (R10). Samples were run in triplicate. Previous experiments had shown that there was an explosive burst of p24 production between 30 and 60 h of culture, which coincided with visible cytopathic effects on the cells (ballooning, syncytia formation, blebbing, and cell death). Thus, supernatant samples were taken during this interval. For testing PHA blasts, PBMCs from a healthy donor were collected and purified by ficoll gradient and cryopreserved according to standard procedures. Subsequently, 1 × 10^8^ thawed PBMCs were cultured in R10 media with 5 μg/mL PHA for 2 days at 37 °C. PBMCs were then washed with R10 twice and resuspended in 5ml R10. Cells were infected with 1ml of HIV_BAL_ virus equivalent to 10^5^ TCID_50_ in the presence of IL-2 and incubated overnight. Cells were washed twice with R10, placed in a 48 well plate (2 × 10^6^/mL), and either IgG, J3, or J3ExoA were added at 50nM. Plates incubated at 37 °C for 5 days. Supernatant samples were assayed for the production of HIV capsid protein, p24, using an antigen capture enzyme-linked immunoassay (ELISA) that we have described in detail elsewhere [60]. Standard curves utilizing purified p24 antigen (HIV Reagent Program) were performed on each ELISA plate allowing for accurate quantitation of supernatant p24. To measure the abilities of J3ExoA and J3VHH to inhibit the binding of HIV to cells, dilutions of J3VHH and J3ExoA were mixed with a pre-defined concentration of HIV_AD8_, and after 1 h transferred to MAGI-CCR5 cells (NIH HIV Reagent Program, ARP-3522, contributed by Dr. Julie Overbaugh) [61]. Production of β-galactosidase was assayed at 48 h and results presented as percent neutralization normalized to untreated virus.

### 4.6. Mice and Immunizations

The Animal Resource Center at Montana State University, the site of the animal studies is an AALAS-approved laboratory facility, meeting all federal and state requirements for animal care. The experimental protocol (2017-139) was approved by the Montana State University Institutional Animal Care and Use Committee. Female and male mice were caged separately with a maximum of five animals per cage. BALB/c mice were obtained from Jackson Labs (Bar Harbor, ME, USA). Groups of 6-week-old 8 mice (4 female, 4 male) were bled, via mandibular or tail vein, and then injected intraperitoneally with 10 µg of IT diluted in PBS to 100 µL. Endotoxin contamination of the immunogens met standards for human use. Mice then received three additional weekly injections, followed by four injections every two weeks. Mice were bled prior to injections on weeks 2, 5, 9, and two weeks following the final injection. Mice were followed closely for clinical signs of distress as detailed in the experimental protocol. There was no evidence of pain or distress in the mice. Mice were sacrificed by CO_2_ asphyxiation followed by thoracotomy.

### 4.7. Enzyme Linked Immunoassays

The production of antibody to the ITs and their components was measured by ELISA using established protocols [13,62]. Intact IT (J3ExoA or 7B2-dgA) or components (J3VHH or PE24) at 1 µg/mL in PBS were used to coat wells of Immulon 2HB plates (Thermo, Waltham, MA, USA). Following 18–24 h at 4 °C, the plates were blocked with PBA for a minimum of 18 h. Immediately prior to use, the plates were washed with PBS. In the experiments shown here, the plasma from all mice in each experimental group was pooled and diluted into PBA. The plasma was plated in triplicate into the antigen-coated wells, and incubated overnight at 4 °C. The plates were then washed 6× with PBS/1% Tween 20 (Sigma) and alkaline phosphatase-conjugated secondary antibody (goat anti-mouse IgG H+L, Invitrogen, Waltham, MA, USA) added at 2 µg/mL in PBA. Following an additional 24 h incubation at 4 °C, plates were washed 5× in PBA with a final wash in PBS. Substrate (BluePhos Microwell Substrate Kit, Seracare, Gaithersburg, MD, USA) was added to wells. Plates were read on a microplate reader (BioTek, Winooski, VT, USA) at 650 nM 30–40 min following the addition of substrate.

### 4.8. Data Display and Statistical Analyses

Samples were run in triplicate. Graphs show mean and standard errors. When no error bars are visible, they are smaller than the symbol on the graph. We show results with J3ExoA in orange and 7B2-dgA in blue. Calculations were performed and graphs produced using GraphPad Prism v10 (GraphPad, Boston, MA, USA). Analyses to determine statistical significance were performed with either a two-sample, two-tailed, unpaired student’s *t* test or one-way ANOVA employing Tukey’s multiple comparison test, whichever was appropriate.

## 5. Conclusions

We have produced an anti-HIV immunotoxin, J3ExoA, that kills HIV-infected cells and suppresses infection. This immunotoxin was created by genetically fusing a llama-derived anti-gp120 VHH to a genetically modified *Pseudomonas* exotoxin A. There was no evidence of toxicity in mice. This immunotoxin has potential for use in “activate and purge” protocols for the eradication of the reservoir of HIV infection that persists despite years of highly effective antiviral therapy.

## 6. Patents

United States Patent 10,010,625, R. Markham, et al. July 3, 2018, Antimicrobial compositions comprising single domain antibodies and pseudomonas exotoxin.

## Figures and Tables

**Figure 1 pharmaceuticals-18-01305-f001:**
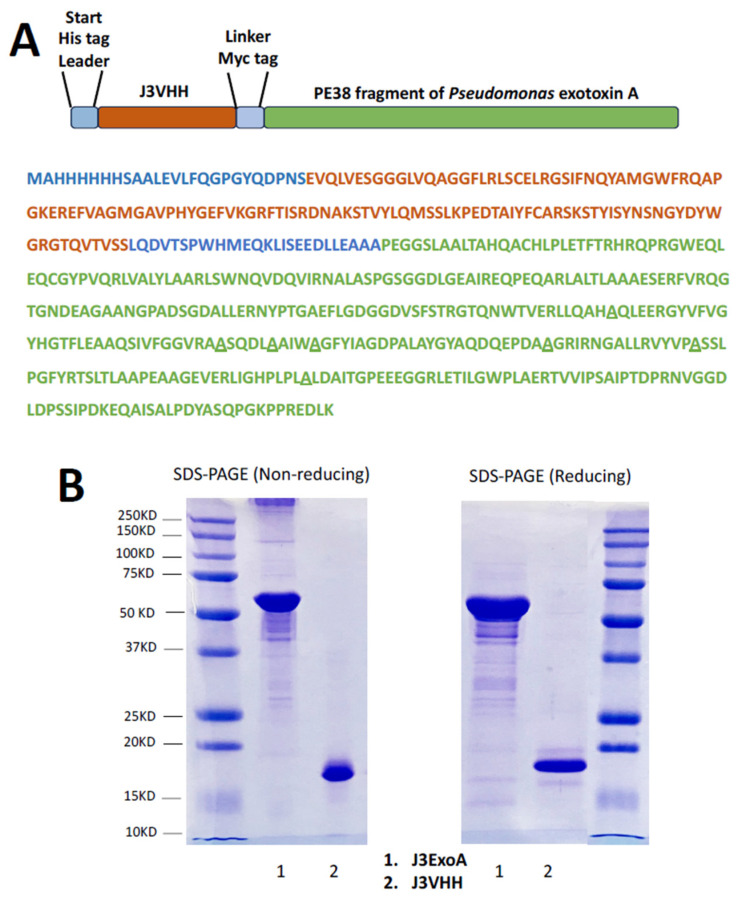
Structure of J3ExoA. (**A**) Stick figure and color-coded amino acid sequence of J3ExoA. The amino acid sequence of J3VHH is shown in light brown, ExoA in green. Underlines beneath alanines indicate that these are introduced alanine substitutions to reduce immunogenicity. (**B**) Confirmation of molecular weights of constructs. Samples were denatured in SDS at 95°, run on a 5–15% polyacrylamide gradient gel under reducing and non-reducing conditions. Each gel shows size markers.

**Figure 2 pharmaceuticals-18-01305-f002:**
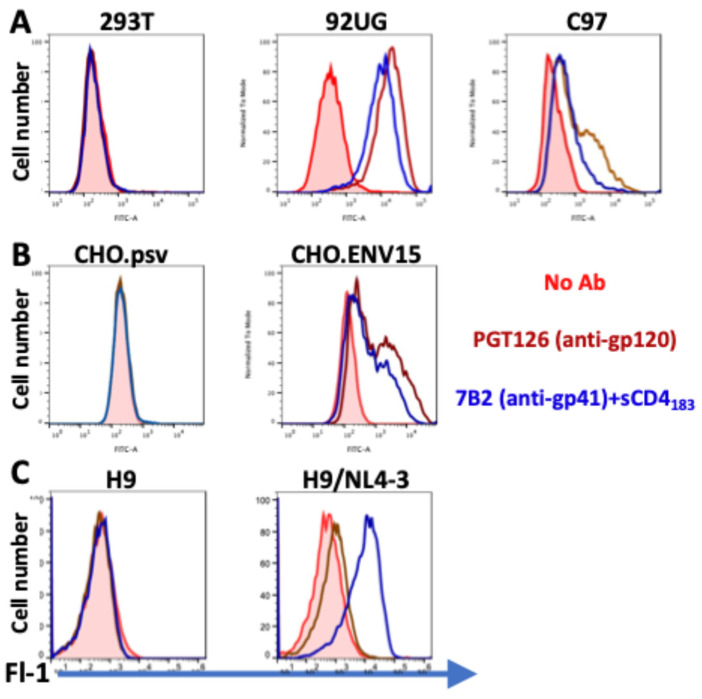
Expression of the HIV envelope on the surface of target cells. Cells were incubated with no mAb, anti-gp120 mAb PGT126, or anti-gp41 mAb 7B2+sCD4, washed and stained with FITC-conjugated goat anti-human IgG. Fluorescence was quantified using an Accuri C6 flow cytometer. Fluorescence is shown on the horizontal axis as a log scale, cell number is shown on vertical axis. The Env-negative parental cells are shown in the left column. (**A**) shows HEK-293T transfected cell lines. (**B**) CHO cells. (**C**) uninfected and HIV-infected H9 lymphoma cells.

**Figure 3 pharmaceuticals-18-01305-f003:**
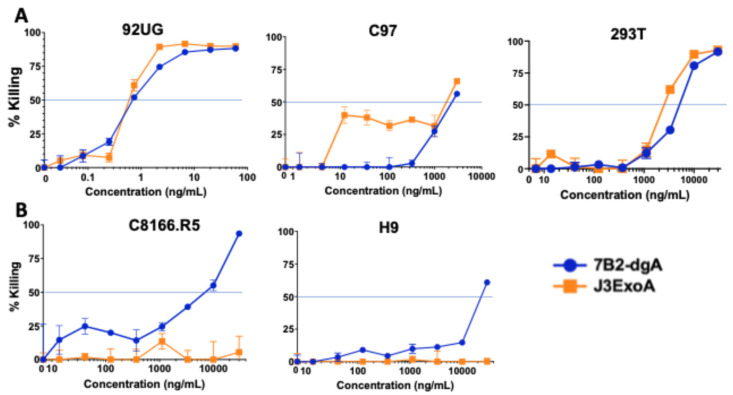
Comparative cytotoxicity of J3ExoA and 7B2-dgA. The indicated target cells were mixed with IT to the final concentration indicated. 7B2-dgA was accompanied by CD4-IgG2 (500 ng/mL) and incubated for 72 h. During the final 3 h, MTS substrate was added, and the optical density of the wells at 490 nm was determined on a microplate reader. The optical density was converted to % killing, which is shown on the vertical axis, with 50% killing marked by the horizontal black line. The concentration is shown on the horizontal axis. Note the marked difference in IT concentrations tested on the Env-negative cell lines. (**A**) shows cytotoxicity on 293T cells, and Env+ cell lines derived from it. (**B**) shows non-specific cytotoxicity on Env-negative CD4+ lymphoma cell lines used in later studies. All samples were run in triplicate, with the mean and SEM shown. If no error bars are visible, then they are smaller than the symbol.

**Figure 4 pharmaceuticals-18-01305-f004:**
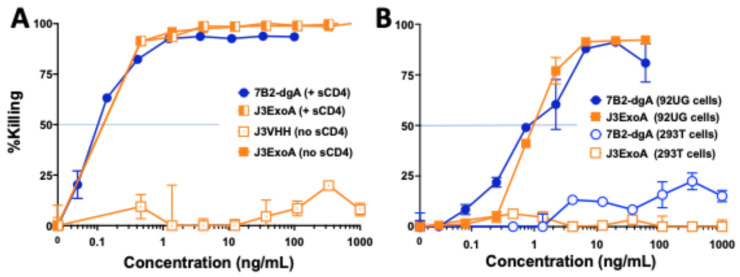
Specificity and effect of sCD4 on IT killing. Assay conditions and data display are as described for Figure 3. (**A**) shows the lack of effect of CD4 on the cytotoxicity of J3ExoA (the two curves overlap) and the absence of killing by J3VHH alone. (**B**) compares the cytotoxicity curves obtained by each IT on the Env+ 92UG cells and Env-negative 293T cells.

**Figure 5 pharmaceuticals-18-01305-f005:**
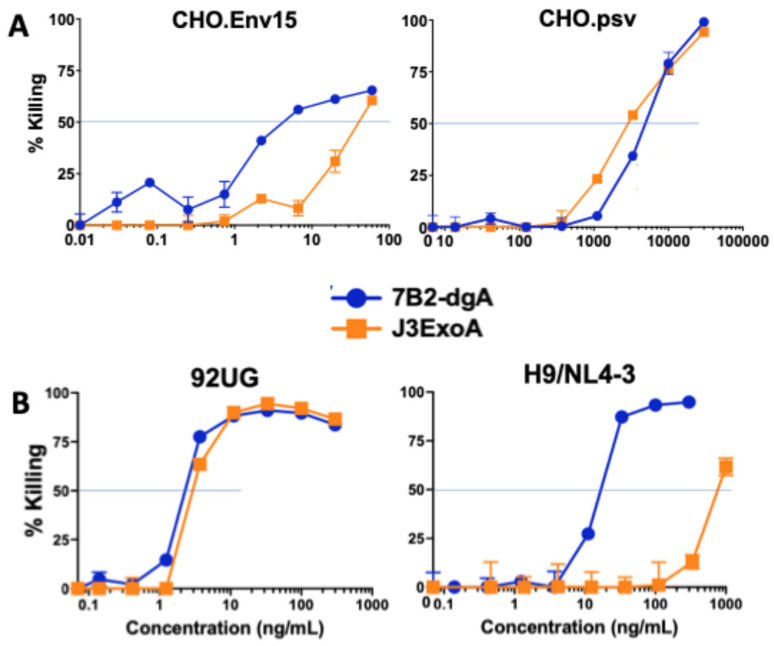
Comparative cytotoxicity of ITs on additional target cell lines. Assay conditions and data display are as described for Figure 3. (**A**) evaluates killing of CHO-derived cells: the Env-negative parent CHO.psv, and the transfected CHO.Env15. (**B**) compares cytotoxicity of the 293T-derived 92UG cells by each IT to killing of the infected T-cell lymphoma H9/NL4-3. Cytotoxicity of H9/NL4-3’s non-infected parent, H9, is shown in Figure 3.

**Figure 6 pharmaceuticals-18-01305-f006:**
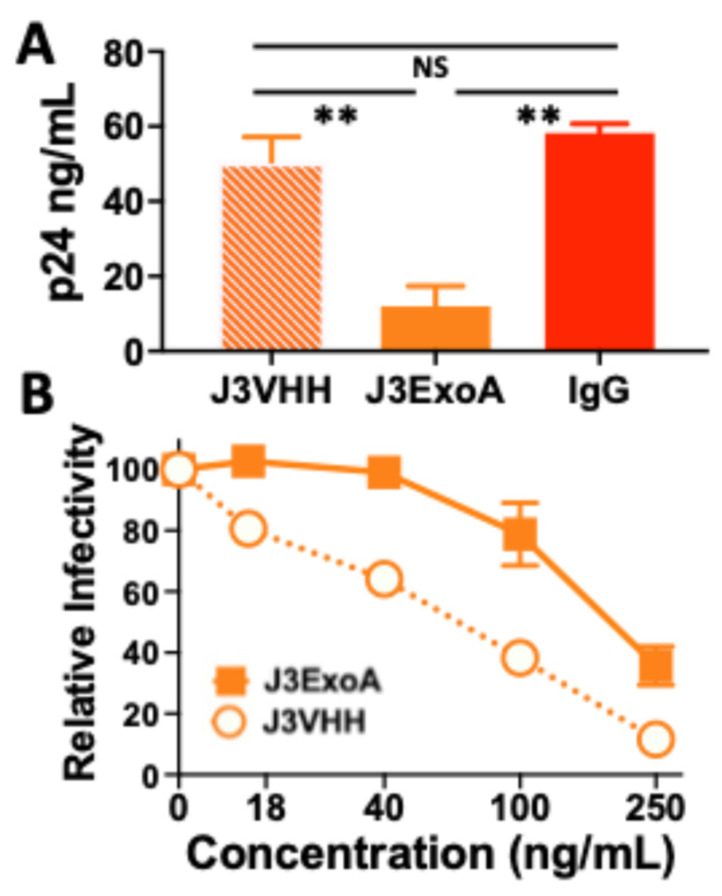
Comparison of anti-HIV activities of J3ExoA and J3VHH. (**A**) PHA blast cultures were infected with HIV_Bal_ and 18 h later 50 nM J3VHH, J3ExoA, or control IgG were added to the culture. Supernatants were tested for p24 antigen five days post-infection. Statistical comparisons were performed using one-way ANOVA employing Tukey’s multiple comparison test, ** *p* < 0.01. (**B**) A pre-defined dilution of HIV_AD8_ was mixed with the indicated concentration of J3VHH or J3ExoA for 1 h and plated onto MAGI-CCR5 cells. Virus infection was measured two days later and results normalized to percent infectivity compared to untreated (0 ng/mL) cells.

**Figure 7 pharmaceuticals-18-01305-f007:**
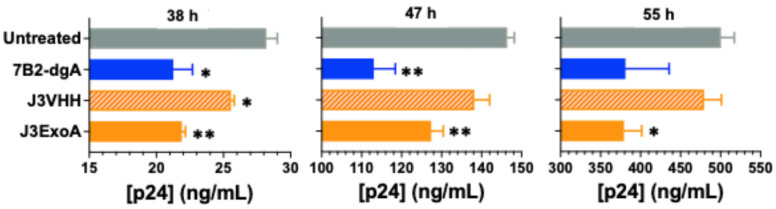
Inhibition of infection of C8166.R5 by ITs. Uninfected C8166.R5 cells were mixed with a limiting dilution of infectious HIV in the presence of IT (50 ng/mL, no sCD4 with 7B2-dgA) HIV p24 antigen was measured at the indicated time points post infection using a standard ELISA. Statistical differences were determined by *t* test, comparing the treatment group to the untreated control (* *p* < 0.05, ** *p* < 0.01).

**Figure 8 pharmaceuticals-18-01305-f008:**
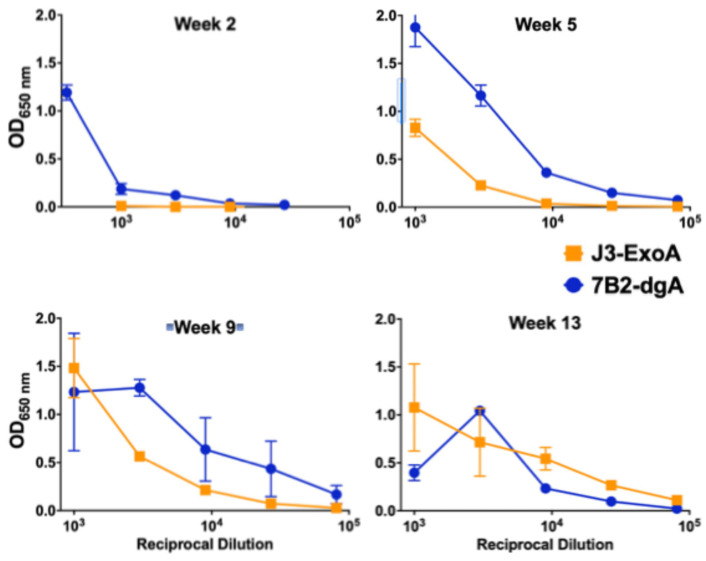
Comparative immunogenicity of J3ExoA and 7B2-dgA measured by ELISA. Groups of eight mice were repeatedly immunized with 10 µg of either J3ExoA or 7B2-dgA. Pooled plasma from the indicated time points were titrated against the IT used to immunize the mice and detected with alkaline phosphatase-conjugated anti-mouse IgG. Each point represents the mean and SEM of triplicate wells. If no error bars visible the error is smaller than the symbol.

**Figure 9 pharmaceuticals-18-01305-f009:**
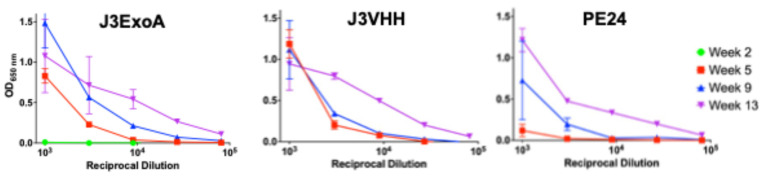
Binding of anti-J3ExoA immune sera to different components of the IT. ELISA plates were coated with 1 µg/mL of J3ExoA, J3VHH, or PE24. Pooled sera of mice immunized with J3ExoA from each time point were titrated for binding to the different IT components as described for Figure 8.

## Data Availability

The data are available from the corresponding authors upon reasonable request.

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
