# Peer review of "J3ExoA: A Novel Anti-HIV Immunotoxin Fusion of Anti-Gp120 J3VHH and PE38 Fragment of Pseudomonas Exotoxin A"

_pharmaceuticals, 2025, doi:10.3390/ph18091305_

Round 1
Reviewer 1 Report
Comments and Suggestions for Authors
Summary of the study:
In this study, the authors designed and produced a novel protein construct by combining an HIV gp120 envelope binding protein (i.e., J3VHH) with the Pseudomonas exotoxin A. For potential application in the “activate and purge” (or “shock-and-kill”) framework of the HIV cure agenda, the resulting J3ExoA chimeric protein was assessed for its ability to kill cells of mammalian cell lines that bear the HIV gp160 envelope proteins (in vitro), inhibit viral replication (in vitro) and its impact on immunogenicity (in vivo). A similar construct (i.e., 7B2-dgA) designed and tested by the authors previously, containing an HIV gp41 envelope binding protein (i.e., 7B2) joined to the ricin plant toxin (dgA), was included as reference. Since the Pseudomonas exotoxin A component of J3ExoA was humanized, the authors hypothesized J3ExoA would be less immunogenic than 7B2-dgA. Both J3ExoA and 7B2-dgA showed specificity in killing HIV-1 gp120/gp41-bearing mammalian cells, with the degree of cell killing being influenced by the levels of HIV-1 gp120/gp41present on the mammalian cells. The reduction in HIV replication in PHA blast cultures by J2ExoA was determined not to be due to the blocking of virus binding by the J3 portion. HIV-1 Infection of C8166.R5 lymphoma cells showed a reduction in HIV-1 p24 levels over time for both J3ExoA and 7B2-dgA, but more persistent for J3ExoA. Unfortunately, although no toxicity was observed in mice, no reduction in immunogenicity of J3ExoA, compared to 7B2-dgA, as observed. The authors concluded that, despite the observed immunogenicity, future work on J2ExoA should be continued.
The work presented in this manuscript is novel. In the absence of a preventative HIV vaccine, there is a need to investigate interventions geared towards HIV remission or functional cure. Protein toxins is a novel approach that aims to “compensate” for the shortfalls of the “activate and purge” approach. Although the authors could not demonstrate a reduced immunogenicity of their J3ExoA construct, it would still be valuable to publish the data presented here.
Minor comments with the study:
- Line 36 – 27, page 1: “Long-lived memory… also contributing.”
Comment: Please include a reference for this statement.
- Lines 40 – 42, page 1 – 2: “We and others… express HIV proteins [9-24].”
Comment: There are 16 publications reference at the end of the sentence, which seems excessive. Given the number of publications referenced here, it would be valuable to provide a paragraph that summarizes the findings of the publications that were references. This will be particularly important for readers not familiar with this specific area of research.
- Line 48, page 2: “VHH nanobodies are single domain…”
Comment: Please define the acronym “VHH” in the text.
- Lines 48 – 50, page 2: “VHH nanobodies… full-sized Abs.”
Comment: Please provide a reference for this statement.
- Lines 50 – 51, page 2: “J3VHH has been… nanobody [26].”
Comment 1: Please indicate which portion of gp120 is targeted by this antibody.
Comment 2: Why is J3VHH mentioned here? What is the relation to 7B2?
- Line 55, page 2: “…if a protein toxin is used, we term it an IT;…”
Comment: Please define the acronym “IT” in the text.
- Line 56, page 2: “is applied if a low MW cytotoxic…”
Comment: Please define the acronym “MW” in the text. Please check with the publisher if this acronym is allowed without defining it.
- Line 56, page 2: “…or radioimmunoconjugate if a high…”
Comment: Is there as acronym for using a radioimmunoconjugate? Please define in the text.
- Lines 63 – 65, page 2: “Seeking to produce an effective… retains toxic activity.”
Comment: In the text, please motivate why the current study did not invlove7B2 conjugated to exotoxin A? Why the switch from 7B2 to J3VHH?
- Line 72, page 2: “ExoA consists of the PE38 fragment of pseudomonas exotoxin A that has been…”
Comment: Please italicize “pseudomonas”.
- Under “2.1. Design and Production of J3ExoA”, page 2.
Comment: In the text, please indicate the sizes (i.e., kDa) of J3VHH and exotoxin A, as well as the expected size of the J3ExoA.
- Figure 1, page 3.
Comment 1: Please add “C” for Celsium after “95o” in the legend.
Comment 2: Please mention in the legend if there was a specific reason why the toxin was not included in the PAGE.
- Line 121, page 4: “…pattern, with neither IT obtaining an IC50 at concentrations <2 µg/mL, equivalent to what…”
Suggested change: “…pattern, with neither IT obtaining an IC50 at concentrations <2 µg/mL, equivalent (for 7B2-dgA) to what…”
- Line 129, page 4: “…H9, the parent of H9/NL4-3. J3ExoA failed to demonstrate cytotoxicity at concentrations…”
Suggested change: “…H9, the parent of H9/NL4-3. J3ExoA failed to demonstrate non-specific cytotoxicity at concentrations…”
- Figure 3, page 5: results for C97.
Comment: C97 is an HIV-1 subtype C envelope. Please comment in the results and discussion whether the observation regarding the incomplete cell killing of C97 env-expressing 293T cells could be related to subtype. This is important since HIV-1 subtype C is a highly prevalent global subtype. What could the implications be for J3ExoA if efficacy is subtype specific?
- Lines 143 – 144, page 5: “Neiter IT achieved… immunofluorescence (Figure 2B).”
Comment: Figure 2 refers to the expression of env on the cells, not the killing of the cells. How is the killing “consistent” with observations related to the levels of env proteins on the cells? Please reword this sentence.
- Line 150, page 5: “…cells, consistent with the lower gp120 expression on those cells.”
Suggested change: “…cells, consistent with the lower gp120 expression on NL4-3 infected H9 cells.”
- Line 172, page 6: “…this assay measures the direct cell-to-cell spread of infection, rather than infection by…”
Comment: Please keep in mind that infection was performed prior to treatment. In this context, the p24 produced (and measure) could be as a result of virus being produced by the already-infected cells. So, the assumption that you measured virus spread through cell-to-cell is not accurate. Please rephrase.
- Lines 276 – 278, page 9: “At the time, cells… days 4 to 5.”
Comment: This sentence describes data (results) that your be reported under the Results section.
- Lines 280 – 281, page 10: “In contrast, J3ExoA significantly inhibited the spread of infection at all time points tested.”
Comment: An increase in p24 was observed for J3ExoA at all time points tested. So “inhibited” is not accurate. Please replace “inhibited” with “reduced” in this sentence.
- Lines 284 – 285, page 10: “The degree to which… clinical benefit.”
Comment: Please include a reference of this statement.
- Under “4.2. Reagents and cell lines”, page 11
Comment: Various HIV isolates, some of different clades, were used in the various assays during the study. Please provide motivation in the text for the selection of each isolate and subtype that was used.
- Under 4.7. Enzyme linked immunoassays”, page 13: “In the experiments shown here, the plasma from all mice in each experimental group was pooled and diluted into PBA.”
Comment: Please motivate in the text why the plasmas were pooled, and p24 ELISAs not performed in the individual plasmas.
- The resolution of Figure 9 should be improved.
Author Response
The work presented in this manuscript is novel. In the absence of a preventative HIV vaccine, there is a need to investigate interventions geared towards HIV remission or functional cure. Protein toxins is a novel approach that aims to “compensate” for the shortfalls of the “activate and purge” approach. Although the authors could not demonstrate a reduced immunogenicity of their J3ExoA construct, it would still be valuable to publish the data presented here.
Minor comments with the study:
- Line 36 – 27, page 1: “Long-lived memory… also contributing.”
Comment: Please include a reference for this statement.
RESPONSE: Done line 36
- Lines 40 – 42, page 1 – 2: “We and others… express HIV proteins [9-24].”
Comment: There are 16 publications reference at the end of the sentence, which seems excessive. Given the number of publications referenced here, it would be valuable to provide a paragraph that summarizes the findings of the publications that were references. This will be particularly important for readers not familiar with this specific area of research.
RESPONSE: Agreed it is a lot of references, but we wanted to include our manuscripts that would be cited later in the paper, as well as acknowledge the work of others. However, we have taken your advice and added a brief discussion (lines 42-45).
- Line 48, page 2: “VHH nanobodies are single domain…”
Comment: Please define the acronym “VHH” in the text.
RESPONSE: Done, lines 51-52
- Lines 48 – 50, page 2: “VHH nanobodies… full-sized Abs.”
Comment: Please provide a reference for this statement.
RESPONSE: Done (line 54)
- Lines 50 – 51, page 2: “J3VHH has been… nanobody [26].”
Comment 1: Please indicate which portion of gp120 is targeted by this antibody.
RESPONSE: Done lines 55-56
Comment 2: Why is J3VHH mentioned here? What is the relation to 7B2?
RESPONSE: J3VHH and 7B2 are mentioned together because they are the targeting domains of the two immunotoxins that are the subject of this manuscript.
- Line 55, page 2: “…if a protein toxin is used, we term it an IT;…”
Comment: Please define the acronym “IT” in the text.
RESPONSE: IT is an abbreviation that is defined in the text (lines 19 and 42) and list of abbreviations.
- Line 56, page 2: “is applied if a low MW cytotoxic…”
Comment: Please define the acronym “MW” in the text. Please check with the publisher if this acronym is allowed without defining it.
RESPONSE: MW is a standard abbreviation for molecular weight and rarely requires a definition.
- Line 56, page 2: “…or radioimmunoconjugate if a high…”
Comment: Is there as acronym for using a radioimmunoconjugate? Please define in the text.
RESPONSE: The abbreviation is RIC, but because this term was used only a few times in the manuscript, we felt it worthwhile to spell it out each time.
- Lines 63 – 65, page 2: “Seeking to produce an effective… retains toxic activity.”
Comment: In the text, please motivate why the current study did not invlove7B2 conjugated to exotoxin A? Why the switch from 7B2 to J3VHH?
RESPONSE: To make a single chain immunotoxin from a standard antibody, such as 7B2, requires that the variable regions of the heavy and light chain antibody are capable of folding into a functional high affinity variable region. On the other hand, J3VHH is already a high affinity broadly neutralizing antibody that exists as a single chain. We have stated this in lines 70-72.
- Line 72, page 2: “ExoA consists of the PE38 fragment of pseudomonas exotoxin A that has been…”
Comment: Please italicize “pseudomonas”.
RESPONSE: Done, line 79
- Under “2.1. Design and Production of J3ExoA”, page 2.
Comment: In the text, please indicate the sizes (i.e., kDa) of J3VHH and exotoxin A, as well as the expected size of the J3ExoA.
RESPONSE: The MW of J3VHH and J3ExoA are now indicated in the text (line 86). The MW of ExoA was not stated because this product alone was never made.
- Figure 1, page 3.
Comment 1: Please add “C” for Celsium after “95o” in the legend.
RESPONSE: Temperatures are routinely given in Centigrade (Celsius), and need not be so indicated. If this journal requires it, the copy editor will indicate so.
Comment 2: Please mention in the legend if there was a specific reason why the toxin was not included in the PAGE.
RESPONSE: As noted in response to #11, the toxin itself was never made. J3ExoA is a genetic construct, not a chemical conjugate.
- Line 121, page 4: “…pattern, with neither IT obtaining an IC50 at concentrations <2 µg/mL, equivalent to what…”
Suggested change: “…pattern, with neither IT obtaining an IC50 at concentrations <2 µg/mL, equivalent (for 7B2-dgA) to what…”
RESPONSE: Actually the statement holds for both ITs, since the concentration required for an IC50 of J3ExoA is the same concentration that gives an IC50 on the Env-negative parent 293T.
- Line 129, page 4: “…H9, the parent of H9/NL4-3. J3ExoA failed to demonstrate cytotoxicity at concentrations…”
Suggested change: “…H9, the parent of H9/NL4-3. J3ExoA failed to demonstrate non-specific cytotoxicity at concentrations…”
RESPONSE: You are right, it reads better (line 137)
- Figure 3, page 5: results for C97.
Comment: C97 is an HIV-1 subtype C envelope. Please comment in the results and discussion whether the observation regarding the incomplete cell killing of C97 env-expressing 293T cells could be related to subtype. This is important since HIV-1 subtype C is a highly prevalent global subtype. What could the implications be for J3ExoA if efficacy is subtype specific?
RESPONSE: J3VHH has been tested for neutralization on a large panel of of viruses including many clade C (see old reference 26 and new reference 3). So we know J3 binds to clade C isolates. The much more likely explanation is that J3ExoA does not kill cells that lack Env or have very low levels of Env. The consistent killing of the ~40% of C97 cells that do express Env is actually a good indication that J3ExoA is effective on this isolate.
- Lines 143 – 144, page 5: “Neiter IT achieved… immunofluorescence (Figure 2B).”
Comment: Figure 2 refers to the expression of env on the cells, not the killing of the cells. How is the killing “consistent” with observations related to the levels of env proteins on the cells? Please reword this sentence.
RESPONSE: In order to kill cells, Env has to be expressed. When the degree of killing that is achieved mirrors the percent of cells expressing Env, then the results can be said to be “consistent”. I have clarified this in the text (lines 152-153).
- Line 150, page 5: “…cells, consistent with the lower gp120 expression on those cells.”
Suggested change: “…cells, consistent with the lower gp120 expression on NL4-3 infected H9 cells.”
RESPONSE: Done (line 159)
- Line 172, page 6: “…this assay measures the direct cell-to-cell spread of infection, rather than infection by…”
Comment: Please keep in mind that infection was performed prior to treatment. In this context, the p24 produced (and measure) could be as a result of virus being produced by the already-infected cells. So, the assumption that you measured virus spread through cell-to-cell is not accurate. Please rephrase.
RESPONSE: I’m not sure I agree with you, because the J3VHH and irrelevant IgG gave similar response. In which case your supposition would argue that there was no spread of infection beyond the initial infection in either culture. In my initial statement I used the word “suggests” to indicate that it was supposition. I have now made the text even more contingent by changing it to “could suggest” (line 180).
- Lines 276 – 278, page 9: “At the time, cells… days 4 to 5.”
Comment: This sentence describes data (results) that your be reported under the Results section.
RESPONSE: This is not really a result, but rather an explanation for how the experiment was done, and why the timing was so important. As a discussion, it belongs exactly where it is.
- Lines 280 – 281, page 10: “In contrast, J3ExoA significantly inhibitedthe spread of infection at all time points tested.”
Comment: An increase in p24 was observed for J3ExoA at all time points tested. So “inhibited” is not accurate. Please replace “inhibited” with “reduced” in this sentence.
RESPONSE: The point has been clarified in the text (line 290).
- Lines 284 – 285, page 10: “The degree to which… clinical benefit.”
Comment: Please include a reference of this statement.
RESPONSE: I’m not sure what should be referenced here, given that this a statement of uncertainty for which no real data exists.
- Under “4.2. Reagents and cell lines”, page 11
Comment: Various HIV isolates, some of different clades, were used in the various assays during the study. Please provide motivation in the text for the selection of each isolate and subtype that was used.
COMMENT: In general using different virus isolates demonstrates that the reagents are broadly reactive and is considered a plus. The cell lines expressing Env have been chosen, quite simply, because they are the only ones we can find that stably express cell surface Env. So we have to take what we can get.
- Under 4.7. Enzyme linked immunoassays”, page 13: “In the experiments shown here, the plasma from all mice in each experimental group was pooled and diluted into PBA.”
Comment: Please motivate in the text why the plasmas were pooled, and p24 ELISAs not performed in the individual plasmas.
RESPONSE: This is a reasonable question to ask. At the start, we did assay individual sera. However, we soon noted that when we took the means of the individual sera and compared that to what was observed when the same sera were pooled, we found no difference. For experimental ease, we decided that moving forward we would test pooled sera.
- The resolution of Figure 9 should be improved.
RESPONSE: Higher quality images were provided directly to the editors. However, when I expanded the image from the manuscript to 2400% I saw no pixilation or blurring, so am unsure what the problem is.
Reviewer 2 Report
Comments and Suggestions for Authors
In this manuscript, the authors continue their work on finding a reagent that can eliminate HIV-infected cells within an infected individual. The idea is to “induce and reduce” the latent reservoir in an individual. The authors had previously designed molecules with good “reducing” activity but felt that their best molecule (7B2-dgA ) was too immunogenic. To that end, they have designed a new bi-specific connecting a gp41 nanobody to a “de-immunized” Pseudomonas exotoxin A molecule. This new molecule (J3ExoA) was shown to have excellent killing activity under different in vitro conditions. However, the authors concluded that the molecule may still be too immunogenic for clinical use.
This is a nice piece of work that follows up on the previous work from this group. Just a few comments are made.
- The in vitro assay system where J3ExoA was not very active was the persistent H9/NL4-3 infected cell line whereas their previous bi-specific, 7B2-dgA, was highly cytotoxic, even though there seemed to be higher levels of gp41than gp120 made in these cells. This begs the question of what the expression levels/cell in this cell line are and how do they compare with the transfected cell lines used in the other assay systems? Also, since there seems to be a certain level of gene expression needed to get cell killing, how does one know whether cells activated with an LRA in vivo will express enough protein for J3ExoA to recognize and kill the activated cells? Please discuss.
- Data in Figure 6: Have the authors examined whether J3VHH reacts with Bal gp41? That could be a simple explanation for the lack of neutralization activity.
- This reviewer thinks that the authors may be giving themselves too high a bar with respect to immunogenicity. One would expect that any foreign (and even autologous) protein administered at high levels to mice would eventually elicit an immune response. Thus, this may not be the best system to evaluate immunogenicity. Using in vitro systems with human helper-T cells (Ito S, Ikuno T, Mishima M, et al. J Immunotoxicol. 2019;16(1):125-132.; Cohen S, Myneni S, Batt A, et al. MAbs.2021;13(1):1898831.) may be a better system to use and may show that these bi-specifics are not as immunogenic in humans as originally thought.
Author Response
This is a nice piece of work that follows up on the previous work from this group. Just a few comments are made.
- The in vitro assay system where J3ExoA was not very active was the persistent H9/NL4-3 infected cell line whereas their previous bi-specific, 7B2-dgA, was highly cytotoxic, even though there seemed to be higher levels of gp41than gp120 made in these cells. This begs the question of what the expression levels/cell in this cell line are and how do they compare with the transfected cell lines used in the other assay systems? Also, since there seems to be a certain level of gene expression needed to get cell killing, how does one know whether cells activated with an LRA in vivo will express enough protein for J3ExoA to recognize and kill the activated cells? Please discuss.
RESPONSE:
Wow, these are important questions that I have pondered myself and am not sure I can give entirely complete answers.
In regard to your first question “what the expression levels/cell in this cell line are and how do they compare with the transfected cell lines”. First I must point out that expression level is only part of the story. The IT must also be internalized. Thus the ratio of cell-surface Env that is internalized vs secreted (eg as virus) also plays a role. We have consistently found that higher concentrations of 7B2-dgA are required to kill the infected H9/NL4-3 than transfected 92UG and have attributed this to secreted virus binding up the IT before it can be internalized. So yes there are differences between infected and transfected cells.
Regarding the second question “how does one know whether cells activated with an LRA in vivo will express enough protein for J3ExoA”. That is the million-dollar question, and as yet there is no answer. We have to do the experiment, ie treat ART-suppressed, LDA-treated people (or macaques) and see if the ITs work. NIH has a current RFA to address the level of Env expression in vivo, so it is very much an open question under active investigation. I have added a statement to the text emphasizing this key unknown (lines 340-341)
- Data in Figure 6: Have the authors examined whether J3VHH reacts with Bal gp41? That could be a simple explanation for the lack of neutralization activity.
RESPONSE:
J3VHH has been reported to neutralize BaL with an IC50 of 0.01-0.1 µg/mL (reference 26, new reference 3). We did not test it, but the efficacy of J3ExoA (figure 6 panel A) suggests that J3 is reactive with the BaL isolate we used.
- This reviewer thinks that the authors may be giving themselves too high a bar with respect to immunogenicity. One would expect that any foreign (and even autologous) protein administered at high levels to mice would eventually elicit an immune response. Thus, this may not be the best system to evaluate immunogenicity. Using in vitro systems with human helper-T cells (Ito S, Ikuno T, Mishima M, et al. J Immunotoxicol. 2019;16(1):125-132.; Cohen S, Myneni S, Batt A, et al. MAbs.2021;13(1):1898831.) may be a better system to use and may show that these bi-specifics are not as immunogenic in humans as originally thought.
RESPONSE: We agree, we set up an extreme case, and it should be noted that at the early time points there was a difference in the antibody response between the two ITs. The helper cell assay is interesting, but in the end it is the anti-IT antibody response that leads to more rapid clearance and loss of activity of the IT.